# Lost in Translation: Physiological Roles of Stored mRNAs in Seed Germination

**DOI:** 10.3390/plants9030347

**Published:** 2020-03-10

**Authors:** Naoto Sano, Loïc Rajjou, Helen M. North

**Affiliations:** Institut Jean-Pierre Bourgin, INRAE, AgroParisTech, Université Paris-Saclay, 78000 Versailles, France; naoto.sano@inrae.fr (N.S.); loic.rajjou@inrae.fr (L.R.)

**Keywords:** seed germination, dormancy, longevity, seed development, RNA-binding proteins, α-amanitin, proteome, translatome

## Abstract

Seeds characteristics such as germination ability, dormancy, and storability/longevity are important traits in agriculture, and various genes have been identified that are involved in its regulation at the transcriptional and post-transcriptional level. A particularity of mature dry seeds is a special mechanism that allows them to accumulate more than 10,000 mRNAs during seed maturation and use them as templates to synthesize proteins during germination. Some of these stored mRNAs are also referred to as long-lived mRNAs because they remain translatable even after seeds have been exposed to long-term stressful conditions. Mature seeds can germinate even in the presence of transcriptional inhibitors, and this ability is acquired in mid-seed development. The type of mRNA that accumulates in seeds is affected by the plant hormone abscisic acid and environmental factors, and most of them accumulate in seeds in the form of monosomes. Release of seed dormancy during after-ripening involves the selective oxidation of stored mRNAs and this prevents translation of proteins that function in the suppression of germination after imbibition. Non-selective oxidation and degradation of stored mRNAs occurs during long-term storage of seeds so that the quality of stored RNAs is linked to the degree of seed deterioration. After seed imbibition, a population of stored mRNAs are selectively loaded into polysomes and the mRNAs, involved in processes such as redox, glycolysis, and protein synthesis, are actively translated for germination.

## 1. Introduction

The control of seed germination is an important factor for successful agricultural production as it affects early seedling growth, preharvest sprouting/dormancy, and longevity (storability). The molecular basis of this regulation has been studied for many years with a landmark discovery made in the 1960s, when *de novo* protein synthesis was observed in germinating cotton seeds even if transcription was inhibited during imbibition [1,2]. This demonstrated that protein synthesis in the early phase of germination uses pre-existing mRNA templates that are stored in mature dry seeds. Some of these ‘stored mRNAs’ are also referred to as ‘long-lived mRNAs’ because they remain translatable for long periods, even if the seeds are exposed to stressful conditions. These findings highlighted the extreme complexity of the molecular processes regulating seed germination (i.e., transcription during seed formation, stability of mRNAs during seed storage/dormancy break, *de novo* transcription after seed imbibition, and translation of these mRNAs at all developmental stages. Thanks to recent advances in post-genomics and multi-omics approaches, facets of the translational regulation of stored mRNAs are gradually being resolved [3,4]. This review outlines the latest findings on stored mRNAs, focusing on their physiological role in seed germination.

## 2. Contribution of Stored mRNAs to Seed Germination

### 2.1. Germination Ability of Mature Seeds Using Stored mRNAs

Visible seed germination (also called germination *sensu stricto*) is often defined as an extension of a part of the embryo (usually the radicle) to penetrate the structures that surround it such as the seed coat [5]. A number of studies have examined whether germination occurs when gene expression is inhibited through seed absorption of exogenously applied drugs. Seed germination was completely blocked in the presence of a cytoplasmic translational inhibitor cycloheximide in lettuce, wheat, Arabidopsis, and rice [6,7,8,9,10], suggesting that *de novo* protein synthesis during imbibition is a prerequisite for germination. RNA polymerase II-mediated transcription can be blocked by α-amanitin and this has been shown to effectively inhibit in vivo transcription during seed imbibition in Arabidopsis and rice, with a significant reduction in UMP incorporation into RNAs synthesized in germinating seeds. Nevertheless, visible germination was observed in the presence of α-amanitin, although germination was retarded [8,11,12]. Rice seeds also showed a similar germination delay with actinomycin D, which intercalates into the DNA template to form a stable complex, thereby inhibiting transcription [9]. These suggest that cytoplasmic translation using stored mRNAs is sufficient for visible seed germination by radicle cell elongation, but that stored mRNAs alone are not enough for normal germination speed. In other words, *de novo* transcription upon imbibition will be required for rapid germination. Nevertheless, lettuce and wheat seeds treated with a transcriptional inhibitor did not show clear germination or embryonic growth [7,13], indicating that germination potential in the presence of inhibitors may differ among plant species or varieties. Additionally, Arabidopsis seed germination was strongly suppressed by a different inhibitor cordycepin (3′-deoxyadenosine) [14], which can be incorporated into RNA and inhibits the elongation of transcripts due to the absence of a hydroxyl group at the 3′ position (Holbein et al., 2009). Nevertheless, cordycepin may also suppress germination through the inhibition of protein synthesis as it blocks the target of rapamycin (mTOR) signaling pathway at higher concentrations [15]. Furthermore, the latter has been associated with abscisic acid (ABA) signaling and growth processes in seeds and plants [16,17]. Moreover, it cannot be ruled out that uptake efficiency into cells may differ depending on the type of drug. Alternatively, as described below, the accumulation of mRNAs during seed formation can be affected by the environmental conditions in which the mother plant is grown, which could conceivably be responsible for variations observed when the transcription of stored mRNAs is inhibited during germination. It is important to note that while germination *sensu stricto* was not suppressed in seeds treated with a transcription inhibitor, subsequent growth was completely blocked (i.e., cotyledon development in Arabidopsis and radicle elongation in rice) [8,9], indicating that *de novo* transcription upon imbibition is indispensable for normal seedling establishment, which requires cell division and differentiation.

### 2.2. Accumulation of Stored mRNAs during Seed Development

Microarray analysis has shown that more than 12,000 and 17,000 different types of stored mRNAs are present in mature dry seeds of Arabidopsis and rice, respectively [18,19]. It is unlikely, however, that all of these stored mRNAs are used in the germination process, as many could be involved in housekeeping activities in cells or persist from seed development processes. Stored mRNAs could be considered as a backup to late seed maturation that take into account the mother plant’s history, adjusting the seed to cope with environmental fluctuations and manage dormancy appropriately, without affecting the ability to germinate. After embryo morphogenesis is completed during seed formation, precocious germination of the embryo can occur if it is removed from immature seed [20,21]. Treating such embryos with a transcriptional inhibitor ascertained when germination-related mRNAs first appear during seed development. In rice, mRNA accumulation was estimated at between 10 to 20 days after flowering (DAF), as actinomycin D inhibited precocious germination of embryos at 10 DAF but not at 20 DAF [22]. RNA-Seq in the same study also detected 529 mRNA candidates unique to germination, which specifically accumulate in embryos at this developmental phase. Similarly, in cotton, germination-specific mRNAs were reported to accumulate between about 30 and 50 DAF [23,24]. Interestingly, young embryos in both rice and cotton exhibited a germination delay compared to mature ones, even without a transcriptional inhibitor. Moreover, during the precocious germination of young embryos, transcription was observed from genes for mRNAs stored for germination (i.e., transcripts of calcium ion and phospholipid signaling-related genes and negative regulators of ABA signaling-related genes increased in young germinating rice embryos [22] as well as inferred increases in transcripts for carboxypeptidase and isocitratase in the cotyledons of young germinating cotton based on enzymatic activities) [23,24]. These imply that some transcription is required in the young embryo during both seed maturation and precocious germination in order to germinate vigorously.

### 2.3. Proteins Encoded by Stored mRNAs are Required for Germination

With low transcriptional activities during the early stages of imbibition, protein synthesis must rely on translation from stored mRNAs in seeds. An effective strategy to comprehensively identify the proteins synthesized from stored mRNAs during germination has been proteome analyses combined with the use of a transcriptional inhibitor [8,9,12,25]. These have shown that 12S seed storage proteins and members of the dehydrin family are translated from stored mRNAs in Arabidopsis seeds [8]. Detailed time course proteomics using [^35^S]-methionine identified similar storage proteins and desiccation tolerance-related proteins as neo-synthesized proteins upon imbibition [26], suggesting that germination begins with a resumption of the seed maturation program through the translation of these mRNAs, and this may be an important checkpoint for seeds to ensure that germination occurs in a favorable environment [3,27]. Recently, proteins translated from stored mRNAs as well as from *de novo* transcribed mRNAs at an initial phase of germination were analyzed in rice. The former proteins were related to glycolysis and translation while the latter were involved in pyruvate metabolism, tricarboxylic acid (TCA) cycle, and momilactone biosynthesis, indicating that upon imbibition, stored mRNAs support the initial production of energy by glycolysis and the activation of translational machinery, whereas *de novo* transcription accelerates energy production after glycolysis, which enables seeds to germinate vigorously [12]. This seems to be reasonable as ATP production by the TCA cycle in mitochondria may not be fully functional during the initial phase of germination. In higher plants, most ATP is provided by oxidative phosphorylation in mitochondria, but the mitochondria in dry seed cells, termed promitochondria, do not have typical cristae structures and must be repaired and undergo differentiation upon imbibition in order to produce sufficient ATP [28,29]. These developed into more typical mitochondria between 12 to 24 h after imbibition in maize and rice [30,31]. It is also noteworthy that during the seed-to-seedling transition in Arabidopsis, the population of translating mRNAs largely overlapped with genes regulated during hypoxia stress [32], implying the importance of translational control under low-oxygen conditions during germination. Interestingly, mRNAs encoding components of the plastid transcriptional machinery were specifically preserved or even newly synthesized during desiccation and stored in dry seeds and could be immediately available on imbibition [33,34].

### 2.4. Selective Translation of Stored mRNAs upon Imbibition

Changes in transcript abundance do not always correlate with observed transitions in protein levels during seed germination [12,26]. One of the mechanisms affecting this is the selective translation of mRNAs involved in germination from over 10,000 types of stored mRNAs. Additionally, the population of transcripts selected for translation has been observed to vary according to the stage within the seed germination program [9,25,26], emphasizing the fact that mRNA translation is both temporal and selective during germination. The latest translatome analyses carried out on Arabidopsis seeds reported that most of the stored mRNAs were detected in the monosome fraction, with translationally quiescent mRNAs associated with single ribosomes, rather than polysomes in which actively translated mRNAs are associated with multiple ribosomes [32,35]. Nevertheless, 17% of monosome specific-stored mRNAs were translationally up-regulated during early seed germination and they encoded proteins involved in processes such as response to water deprivation and cell cycle arrest, which is in agreement with the predicted function of proteins neosynthesized during seed germination [26]. In contrast, non-monosome specific ones (i.e., mRNAs detected from both monosomes and polysomes) were translationally down-regulated during the initial phase of germination [35]. This suggests that the use of stored mRNA for germination involves two step-wise mechanisms: (i) mRNA accumulates as monosomes rather than polysomes during seed formation; and (ii) selective translation of monosomal mRNAs after imbibition. Furthermore, such ribosomal level regulation could allow seeds to pause translation and stack mRNAs, thereby protecting some specific mRNA populations. How specific mRNAs are targeted to monosome complexes and are specifically translated during imbibition is still unknown, however, the same study found clues that monosome specific-stored mRNAs have features such as shorter transcripts, low GC% in UTRs, weak secondary structure, and a motif (GAAGAAGAA) in 5′UTRs [35].

Selective translation can also result from structural features and regulatory sequences within mRNA [36]. The canonical end modifications of seed stored mRNA remain to be fully characterized. Previous work suggests that both cap-dependent and cap-independent translation can occur from stored mRNA to control seed germination. The cap-independent process occurs through the direct recruitment of ribosomal subunits on specific *cis*-acting RNA sequences known as internal ribosome entry sites (IRES) [37]. IRES-specific cellular *trans*-acting factors (ITAF) are important proteins for cap-independent translation initiation. The ErbB3-binding protein 1 (EBP1) is an evolutionarily conserved ITAF found in both animals and plants and previously detected in seeds [38,39]. *EBP1* is highly expressed during germination and the protein is over accumulated during seed priming [39]. Furthermore, in maize embryonic axes, some stored mRNAs were efficiently translated via a cap-independent mechanism during germination [40]. As both stored and *de novo* synthesized mRNAs co-exist during the germination process, it is possible that these different initiation systems allow their selective recruitment during germination. Interestingly, EBP1 is also implicated in stress, and ABA responses [41] and seeds of the Arabidopsis *ebp1* mutant were less sensitive to ABA and germinated more rapidly than those of wild type (WT).

## 3. Stored mRNAs in Seed Dormancy

Seed dormancy is an adaptive trait that determines the timing of germination, thereby allowing seedling establishment under favorable environmental conditions. Most species’ seeds including Arabidopsis acquire a so-called physiological dormancy during seed development, which is controlled by the balance of endogenous hormones [42,43]. ABA is required for the induction of dormancy during seed maturation and gibberellins (GA) for germination upon imbibition. These two hormones negatively influence each other’s biosynthesis and signaling pathways [44]. Physiological dormancy can be released by dry storage (after-ripening) or by hydration under specific moisture, temperature, and light conditions.

### 3.1. Induction of Dormancy and mRNA Accumulation during Seed Maturation

In Arabidopsis, approximately 500 of the most abundant stored mRNAs in dry seeds have been identified, and ACGT-containing ABA-responsive elements (ABREs), CE, and RY *cis*-elements, which are targets of ABA-responsive transcription factors, were highly over-represented in the promoters of the stored mRNA genes [18], suggesting ABA-mediated transcription plays a key role in determining the patterns of the abundant class of stored mRNAs. Environmental factors experienced by the mother plant during seed maturation like temperature, light, and soil nitrate availability also influence the level of dormancy [45], and the accumulation pattern of a subset of stored mRNAs was affected by these environmental conditions [46,47,48]. Nevertheless, transcriptomes in mature, dry Arabidopsis seeds of less-dormant Col and dormant Cvi accessions resemble each other, suggesting that major patterns of stored mRNA accumulation reflect neither the degree of dormancy nor germination potential, but rather the developmental context such as seed maturation [49].

GA biosynthesis is negatively regulated during seed development in order to prevent precocious germination and to allow seed dormancy induction [50]. Accordingly, transcripts of GA 20-oxidase and GA 3-oxidase involved in GA biosynthesis were not detected in the microarray analysis of dry Arabidopsis seeds, although their levels increased after seed imbibition [51], suggesting that mRNAs required for GA biosynthesis are not sufficiently abundant in mature dry seeds to stimulate germination. This was corroborated by a germination test in which a combination of both GA_4+7_ and α-amanitin were applied exogenously, as the addition of GA_4+7_ attenuated the inhibitory effect of α-amanitin on seed germination [8].

### 3.2. Oxidation and Stability of Stored mRNAs during after-Ripening

After maturation, the water contents of dry, orthodox seeds are normally less than 20%, and the cytoplasm of seed cells becomes highly viscous and is transformed into a glass [52]. In the glassy state, cellular components are stabilized and their mobility is severely restricted, preventing enzymatic reactions with free available water. Several studies have shown that active transcription/translation (recruitment of mRNAs to polysomes) does not occur during after-ripening of seeds in a fully dry state [53,54,55,56], raising the question of how dormancy is lost in dry, metabolically inactive seeds. Major molecular events occurring in dry seeds are, therefore, limited to passive and non-enzymatic reactions such as oxidation and Amadori–Maillard reactions. RNA is particularly exposed to oxidation, partly due to its single-stranded structure [57], and it is generally considered that in the absence of efficient repair systems, damaged RNA is enzymatically degraded rather than restored [58]. In fact, oxidation of stored mRNAs to alleviate dormancy has been observed in sunflower during after-ripening [59]. Although ‘passive’, this mRNA oxidation in seeds was not random but highly selective, mainly targeting stored mRNAs encoding proteins involved in cell signaling such as the protein phosphatase 2C PPH1, mitogen-activated protein kinase phosphatase 1, and phenyl ammonia lyase 1. An *in vitro* translation assay also demonstrated that mRNA oxidation caused suppression of protein synthesis with both oxidized and non-oxidized mRNAs associating with ribosomes, but the presence of oxidized bases reducing translation fidelity and increasing the formation of truncated proteins [60]. Similar selective mRNA oxidation was found in after-ripened wheat seeds, although none of the oxidized mRNAs corresponded to a gene orthologue of those identified in sunflower, implying that the mRNAs targeted for after-ripening-induced oxidation are distinct between species [61,62]. These suggest that seed dormancy release by dry after-ripening is associated with targeted mRNA oxidation, which could lead to translational repression of the corresponding proteins during imbibition.

Even in the absence of active transcription and enzymatic degradation during dry after-ripening, the observed changes in the relative amounts of some stored mRNAs in comparative transcriptome analysis may be due to differences in their stability. In fact, some stored mRNAs showed changes in relative transcript abundance during after-ripening in Arabidopsis [63]. This difference in stored mRNA level was correlated with previously reported mRNA stability, where about 13,000 transcripts were analyzed by measuring transcriptome changes in cell cultures after treating with a transcriptional inhibitor [64] (i.e., stable mRNAs appeared to be transcriptionally upregulated due to proportional increases caused by the decreased abundance of unstable ones). Moreover, two positive regulators of seed dormancy, DELLA *GAI* (*GA-INSENSITIVE*) and the histone deacetylase *HDA6/SIL1* exhibit strong transcriptional down-regulation during after-ripening, suggesting that the level of dormancy-promoting stored mRNAs is specifically reduced during after-ripening, thereby increasing germination potential [63].

### 3.3. Fine-Tuning of Stored mRNA Levels after Imbibition Affects Dormancy

The fact that RNA turnover controls the depth of dormancy was more directly corroborated by the analysis of imbibed seeds of Arabidopsis mutants *varicose* (*vcs*) and *exoribonuclease4* (*xrn4*), which are affected in the 5′–3′ mRNA decay process [65]. Seeds of *vcs* mutants were less dormant than WT seeds, being consistent with changes in transcript levels, in which dormancy related mRNAs (ABA anabolism and response, GA inactivation) were suppressed but germination related mRNAs (ABA catabolism, GA activation, and response) were up-regulated. In contrast, *xrn* seeds were more dormant with opposite patterns of gene expression, indicating that on imbibition, VCS and XRN are involved in the turnover of specific subsets of mRNA including those related to ABA and GA, and thereby regulate dormancy. The transcripts putatively degraded by XRN4 or VCS during seed imbibition could be distinguished by distinctive features in their 5′UTR such as a high GC content, strong secondary structure, and specific motifs.

Several pioneering studies on translatome profiling have highlighted the importance of translational control in dormancy after imbibition [32,55,56]. In both dormant and non-dormant dry seeds, monosomes associated with stored mRNAs were abundant while polysomes were scarce or absent. Upon imbibition, there was a transient decrease in monosomes associated with a concomitant increase in polysomal mRNAs, suggesting that the translational activity of stored mRNAs was similar in both kinds of seeds. Nevertheless, this similarity was only observed at the onset of imbibition as non-dormant seeds contained more free monosomes after 24 h of imbibition. This implies that the storage of ribosomes in an unproductive state in non-dormant seeds might serve as a reserve of active ribosomes to kick-start active translation following radicle protrusion [56]. Nevertheless, specific subsets of polysomal mRNAs were detected in dormant or non-dormant seeds upon imbibition, and these were associated with functions such as transport, regulation of transcription, cell wall modifications [55], redox and lipid metabolism [56], and tryptophan-dependent auxin biosynthesis [14]. It is worth noting that transcriptomics highlighted a link between dormancy release and the accumulation of mRNAs associated with the protein synthesis machinery, which led to the concept that translational capacity plays a major role in the switch from dormant to non-dormant state [66]. Such translational control for breaking dormancy may be, at least partially, controlled by GA signaling as after-ripening and subsequent cold stratification resulted in massive mRNA increases for translation-associated genes in WT, but not in the GA-insensitive *sly*1-2 mutant [67]. Similarly, during dormancy release in response to cold stratification or exogenous nitrate, the accumulation of several proteins involved in translational machinery has been observed, suggesting that either the turnover of these specific proteins is affected when seeds are maintained in a dormant state or that the synthesis/stability of the corresponding mRNAs is enhanced upon dormancy release [68]. Transcripts encoding translational machinery were also accumulated at non-dormant stages during dormancy cycling in the field [69]. Together, these observations suggest that some mRNAs are selectively incorporated into polysomes during imbibition in a manner that is dependent on the depth of dormancy, and that this is part of the biological process that regulates dormancy. Distinctive 5′UTR features such as GC content and the number of upstream open reading frames could play a role in this selective translation [56], although the exact molecular basis for such selectivity is still unknown. Further insight into the function of these RNA features may be gained by mutation (i.e., of uORF by CRISPR/cas9 in plants) [70].

## 4. Stored mRNAs in Seed Longevity

‘Seed longevity’, defined as the total time span during which seeds remain viable, is an important trait for ecology, agronomy, and economy [71]. During dry storage, seed viability gradually decreases due to ‘aging processes’ and/or ‘deterioration events’. The molecular aspects of seed longevity have been reviewed [71,72,73,74]. In brief, reduction of longevity during dry seed storage is mainly caused by oxidation of cellular macromolecules such as nucleic acids, proteins, and lipids because enzymatic reactions are restricted by the lack of free available water, similar to the process of dry after-ripening described above (Section 3.2).

### 4.1. Total RNA Degradation during Seed Aging

The degradation and fragmentation of 18S and 25S rRNAs in seeds during storage was observed in rye by gel electrophoresis [75], and a similar decrease in rRNA quality was reported in *Nicotiana tabacum* L. [76]. A positive correlation between total RNA amounts (or 18S and 25S rRNAs) and the germination ability of aging seeds was also found in carrot and sunflower [77,78], suggesting that the rRNA quantity and quality decreases as seeds deteriorate. Additionally, a significant reduction in total RNA content and integrity was previously detected in aged pea seeds that had only lost 13% of their initial viability [79], implying that changes in total RNA are very early indicators of the onset of seed deterioration. This is consistent with observations in aged Arabidopsis seeds where translational activity declined with the loss of germination potential [80]. Unfortunately, the quality of RNA is difficult to compare between one lab and another (i.e., in general, RNA is subjectively considered as high quality when the ratio of 25S:18S bands is about 2.0 and higher) following gel electrophoresis, and the resulting data cannot be processed digitally.

To estimate the integrity of RNA samples in an unambiguous way, Agilent Technologies have developed a technique that calculates an RNA integrity number (RIN), based on nine features of the electropherogram such as the precursor-region of 25S, 25S, and 18S peaks, and 5S region (small rRNA fragments), generated by micro-capillary electrophoresis of total RNA [81]. The RIN is an index with a value in the range one to 10 (fully degraded RNA to intact RNA) and RNA-Seq protocols typically require total RNA with a value of at least eight [82]. Analyses of seed longevity carried out using the RIN as an index in pea [83,84], soybean [85,86], carrot, crimson clover, lettuce, onion, safflower, sesame, sorghum, tomato, and watermelon [84], revealed a positive and significant correlation between seed viabilities and RIN values in most species. Moreover, the RIN decreased prior to the loss of seed viability. Exceptions were watermelon, where the proportion of germination correlated poorly with storage time, and tomato, which showed electropherogram anomalies that impaired the accurate calculation of RIN. Nevertheless, the integrity of total RNAs in seeds appears to be a good indicator of seed aging that is broadly applicable across species. The link between RNA integrity and seed quality seems reasonable because, as above-mentioned, RNA is more vulnerable to oxidation than DNA due to its single-stranded structure. Among more than 20 different types of base damage caused by hydroxyl radicals, the most prevalent oxidized base in RNA is 8-oxoguanine [57]. Nonetheless, no significant correlation was observed between seed storage time and the relative abundance of 8-oxoguanine in total RNA [86], suggesting that RNA damage occurs by random oxidative attack on the molecule.

### 4.2. Transcriptome Profiles in Aged/Deteriorated Seeds

The transcriptomes of aged or deteriorated seeds including stored mRNAs have been analyzed in order to understand how gene expression contributes to the loss of seed longevity. The latest discovery is that aging soybean seed transcripts are broken non-specifically by free radical attack at random bases and that greater fragmentation occurs in longer transcripts [86]. Many of the shortest (<1200 bp) transcripts were relatively intact even after long-term seed storage and were mainly involved in ribosomal and translational functions, underlining the importance of stored mRNAs in the reconstitution of the translational machinery for germination. The relatively longer transcripts (>2500 bp) were regarded as candidate markers for aging before seeds lose viability, as they were less degraded on short-term compared to long-term seed storage; many of these encoded proteins with ATP-binding functions. Other transcriptomic evidence, that stored mRNAs may be involved in seed longevity, has arrived from beyond the Earth. Rice seeds that were exposed to the environment outside the International Space Station showed lower viability than ground-stored seeds, and the levels of stored mRNAs required for germination, especially involved in glycolysis pathways, had decreased two-fold in space-stored seeds compared to ground-stored ones [87].

As the evaluation of seed longevity during natural aging takes a long time, ‘controlled deterioration treatment (CDT)’ or ‘accelerated aging’, in which seeds are placed under high relative humidity and high temperature, have been used to accelerate seed deterioration as well as to compare transcriptomes in deteriorating seeds [83,88,89]. Many of the differentially expressed genes reported were from the up-accumulation of transcripts during CDT, perhaps because the cells within the seed are no longer in a glassy state due to the high humidity and temperature, which would facilitate the occurrence of cellular processes such as *de novo* transcription [83]. Nevertheless, transcripts whose abundance was reduced during CDT were related to programmed cell death, antioxidants, seed storage proteins, heat shock transcription factors, and the glycolytic pathway [83,88,89], implying the importance of these stored mRNAs for longevity. In both natural and artificial conditions, special attention must be given to the normalization of transcriptomes between samples as well as the selection of appropriate reference genes for qPCR analyses, as RNA integrity in aged seeds will be more or less impaired. To address this, pea RNA was spiked with total human RNA and a human-specific gene was then used as an artificial reference gene for qPCR analysis of aging pea seeds [83]. In addition, a novel quantitative method has been developed to estimate the relative amount of intact stored mRNAs and rate of mRNA degradation at the single nucleotide level based on ΔCt values from qPCR analysis [90]. Additionally, it is not easy to judge whether the seeds used are alive or dead in the dry state, since dried seeds are in a quiescent state where almost all their vital activities have been stopped. Prominent signs of seed aging appear during water-uptake for germination, but it can be costly for the seed industry and seed banks to monitor viability [84]. Degradation of stored mRNA holds the promise of being a useful indicator for predicting the loss of seed life and vigor, even though the molecular mechanisms involved in the protection and repair of damaged mRNAs remain to be elucidated.

## 5. Concluding Remarks and Future Perspectives

Seed germination is a physiologically onerous process in which extremely dry embryos absorb water and grow rapidly to be seedlings. Germination tests of immature seeds with a transcription inhibitor showed that stored mRNAs provide impetus for rapid germination after imbibition. Notably, under hypoxic conditions during early imbibition proteins can be synthesized from stored mRNAs for processes such as redox reactions, energy production through the glycolysis/TCA cycle, and translational regulation. The release of seed dormancy involves the deterioration of specific subsets of stored mRNAs during after-ripening and selective recruitment of the mRNAs to polysomes upon imbibition. Longer-term seed storage causes a loss of longevity that appears to be linked to a decrease in stored RNA integrity, in which shorter transcripts are more likely to remain undamaged (Figure 1).

In order for stored mRNAs to be translated immediately after imbibition, the relevant translational machinery must also be stored in the seed as protein and how the latter is maintained in the dry seed is a key issue to be solved in the future. Moreover, in addition to desiccation, seeds undergo a variety of other stresses in the natural environment. It has been postulated that mRNAs are degraded one after another in seeds that have undergone such stress upon imbibition, and the expression of genes essential for the germination process may be insufficient from transcription alone after imbibition [91]. In other words, seeds may be more apt to respond to these stresses by accumulating a range of mRNAs in advance. Future research is also needed concerning the selective translation of stored mRNA from seeds exposed to stressful environments.

It is well established that regulation of RNA function involves RNA-binding proteins (RBPs) that interact with RNAs and form ribonucleoprotein complexes (i.e., RBPs play an important role in nuclear RNA processing as well as for cytoplasmic RNA transport, localization, stability, and translation in eukaryotes) [92]. Various RBPs have been identified in seeds by proteome analyses [35,93,94,95], and some of these RBPs may interact with specific sequence motifs in stored mRNAs, thereby regulating their localization, stability, or translation. Recently, many research tools have been developed to examine the functions of translation control and RBPs [96,97]. These techniques will further advance the study of stored mRNAs. Stored mRNAs in seeds appear to be “lost” as their translation is temporarily interrupted, but in fact they play important roles in the regulation of seed germination, dormancy and storability.

## Figures and Tables

**Figure 1 plants-09-00347-f001:**
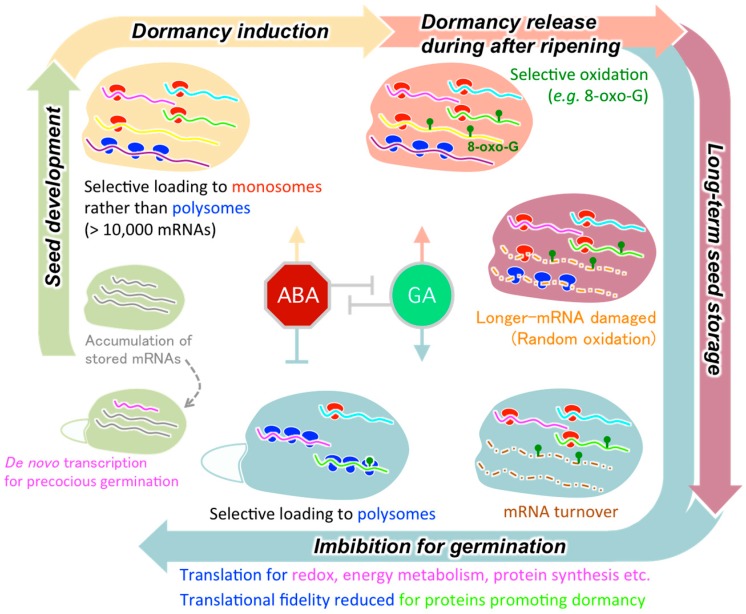
Overview of the regulatory mechanisms controlling stored mRNA translation during germination, dormancy, and longevity in the plant life cycle from seed development to germination. See main text for detailed explanation of different mechanisms. ABA, abscisic acid; GA, gibberellins; 8-oxo-G, 8-oxoguanine.

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
