# Peer review of "Lost in Translation: Physiological Roles of Stored mRNAs in Seed Germination"

_plants, 2020, doi:10.3390/plants9030347_

Round 1
Reviewer 1 Report
This review "Lost in translation" (nice title!) is very useful indeed. It is written in a clear and fancy way so that many readers can profit from it.
I certainly recommend publication!
Author Response
We thank reviewer 1 for their encouraging evaluation of our manuscript and for their time spent on reviewing.
Reviewer 2 Report
The authors present here a review that outlines the latest findings on stored mRNAs, focusing on their physiological role in seed germination.
The control of seed germination is an important factor for successful agricultural production.
The authors divided the review in several chapter very well documented and sufficiently described.
I accept the article in the present form.
Author Response
We thank reviewer 1 for their time spent on reviewing our manuscript. The author (HN) who is a native English speaker has carefully reread the text and corrected a few minor spelling and grammatical errors.
Reviewer 3 Report
The authors relate the RNA to the processes associated with abyssic acid. In addition, they indicate their relationship with the redox processes, glycolysis and protein genesis. Many publications have studied the relationship between dormancy and the action of gibberellins is evident and they demonstrate the positive action of gibberelins. For this reason, it might be interesting that the authors deepen a little more in the relationship between gibberellins and the processes described in this paper.
Author Response
We thank the reviewer for dedicating time to reviewing our manuscript and for their helpful feedback. We have now added text and corresponding references in two places (p5, lines 195 to 202; p6, lines 263 to 269) in the manuscript to provide more insight into the relationship between gibberellins and stored mRNAs in seeds. We have also modified the figure to include the role of gibberellins. The author (HN) who is a native English speaker has carefully reread the manuscript to identify and correct the few spelling and grammatical errors that were present.